# Human-Robot Commensality: Bite Timing Prediction for Robot-Assisted Feeding in Groups

**Jan Ondras***
Cornell University
janko@cs.cornell.edu

**Abrar Anwar***
University of Southern California
abrar.anwar@usc.edu

**Tong Wu***
Rutgers University
tong.wu96@rutgers.edu

**Fanjun Bu**
Cornell Tech
fb266@cornell.edu

**Malte Jung**
Cornell University
mfj28@cornell.edu

**Jorge Jose Ortiz**
Rutgers University
jorge.ortiz@rutgers.edu

**Tapomayukh Bhattacharjee**
Cornell University
tapomayukh@cornell.edu

**Abstract:** We develop data-driven models to predict when a robot should feed during social dining scenarios. Being able to eat independently with friends and family is considered one of the most memorable and important activities for people with mobility limitations. While existing robotic systems for feeding people with mobility limitations focus on solitary dining, commensality, the act of eating together, is often the practice of choice. Sharing meals with others introduces the problem of socially appropriate bite timing for a robot, i.e. the appropriate timing for the robot to feed without disrupting the social dynamics of a shared meal. Our key insight is that bite timing strategies that take into account the delicate balance of social cues can lead to seamless interactions during robot-assisted feeding in a social dining scenario. We approach this problem by collecting a Human-Human Commensality Dataset (HHCD) containing 30 groups of three people eating together. We use this dataset to analyze human-human commensality behaviors and develop bite timing prediction models in social dining scenarios. We also transfer these models to human-robot commensality scenarios. Our user studies show that prediction improves when our algorithm uses multimodal social signaling cues between diners to model bite timing. The HHCD dataset, videos of user studies, and code are available at https://emprise.cs.cornell.edu/hrcom/

**Keywords:** Multimodal Learning, HRI, Assistive Robotics, Group Dynamics

## 1 Introduction

Nearly 27% of people living in the United States have a disability, and close to 24 million people aged 18 years or older need assistance with activities of daily living (ADL) [1]. Key among these activities is *feeding*, which is both time-consuming for the caregiver, and challenging for the care recipient (patient) to accept socially [2]. Indeed, needing help with one or more ADLs is the most cited reason for moving to assisted or institutionalized living [3, 4]. Although there are several automated feeding systems on the market [5–13], they have lacked widespread acceptance. One of the key reasons is that all of them require manual triggering of bite timing by the user, which is challenging for users with cognitive disabilities and inconvenient in social settings. A key challenge for the realization of autonomous robotic feeding systems is therefore to infer proper bite timing [14].

While existing systems focus on solitary dining (e.g. [15–32]), **commensality**, the act of eating together, is often the practice of choice. People like to share meals with others. The social experience of a shared meal is an important part of the overall eating experience and current robot feeding systems are not designed with that experience in mind. Transferring the challenge of inferring appropriate bite timing to a social dining setting requires not only attuning to the user's eating

---

*These authors contributed equally to this work.

6th Conference on Robot Learning (CoRL 2022), Auckland, New Zealand.

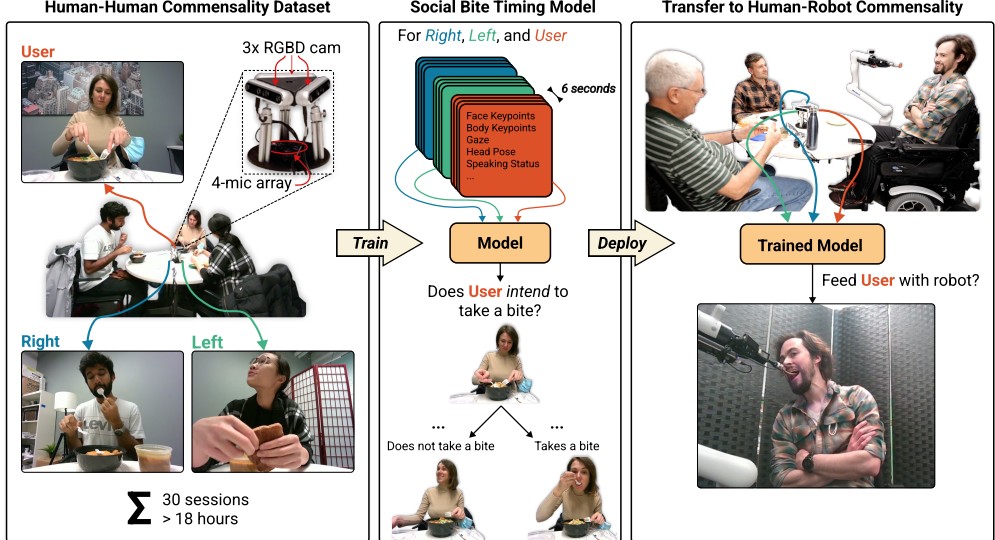

Figure 1: Our bite timing prediction workflow: **(Left)** Human-Human Commensality Dataset collection: We record audio and video of participants eating food in triads. **(Middle)** Our Social Nibbling NETwork (SoNNET) learns to predict whether a user intends to take a bite based on a 6-second window of social signals. **(Right)** We conduct a social robot-assisted feeding user study by deploying a variation of SoNNET on a robot. We refer to the **User** also as a **Target user**.

behavior but also to the complex social dynamics of the group. For example, a robot should not attempt to feed a user who is actively engaged in conversation. Motivated by a growing body of research that seeks to develop models for robots to function in group settings [33, 34] we ask the seemingly simple question: *How should an assistive feeding robot decide the right timing for feeding a user in ever-changing and dynamic social dining scenarios?*

We developed an intelligent autonomous robot-assisted feeding system that uses multimodal sensing to feed people in dynamic social dining scenarios. We collected a novel audio-visual Human-Human Commensality Dataset (HHCD) capturing human social eating behaviors. Using this data, we then trained multimodal machine learning models to predict bite timing in human-human commensality. We explored how our models trained on human-human commensality scenarios performed in a human-robot commensality setting and evaluated them in a user study. The overall workflow is shown in Fig. 1. We made algorithmic and experimental design decisions by consulting with care recipients, caregivers, and occupational therapists. We find that bite timing prediction improves when our model accounts for social signaling among diners, and such a model is preferred over a manual trigger and a fixed-interval trigger. Our main contributions include:

- A SOcial Nibbling NETwork (SoNNET) which captures the subtle inter-personal social dynamics in human-human and human-robot groups for predicting bite timing in social-dining scenarios.
- Methods that can successfully transfer bite timing strategies learned from human-human commensality cues to human-robot commensality situations, which we evaluate in a user study with a robot in 10 triadic human groups.
- A socially-aware robot-assisted feeding system that extends our capacity to feed people in solitary settings to groups of people sharing a meal.
- An analysis of various social and functional factors that affect human feeding behaviors during human-human commensality.
- A novel *Human-Human Commensality Dataset (HHCD)* containing multi-view RGBD video and directional audio recordings capturing 30 groups of three people sharing a meal.

## 2 Human-Robot Commensality

Eating is a complex activity that requires the sensitive coordination of several motor and sensory functions. Anyone who has fed another knows that feeding, particularly *social feeding* where a person is being fed in a social setting, is a delicate *dance* of multimodal signaling (via gaze, facial expressions, gestures, and speech, to name a few). Research on *commensality*, the practice of eating together, has highlighted the importance of the social nature of eating for social communion, order,

health, and well-being [35]. As a consequence, digital commensality has focused on understanding the role of technology in facilitating or inhibiting the more pleasurable social aspects of dining [36].

When a person relies on assisted feeding, meals require that patient and caregiver coordinate their behavior [37]. To achieve this subtle cooperation, the people involved must be able to initiate, perceive, and interpret each other's verbal and non-verbal behavior. The main responsibility for this cooperation lies with caregivers, whose experiences, educational background, and personal beliefs may influence the course of the mealtime [38]. Our goal in this work is to understand the rhythm and timing of this *dance* to enable an automated feeding assistant to be thoughtful of when it should feed the user in social dining settings. We introduce the concept of **Human-Robot Commensality** at the intersection of commensality and robot-assisted feeding in social group settings.

Our research is motivated by the key insight that bite timing strategies that take into account ever-changing social signals and group dynamics can lead to a seamless human-robot collaboration in social dining scenarios. Fueled by this insight, we believe a feeding device that takes the initiative and offers bites proactively during the meal at times when a bite is likely to be desired will create a more seamless dining experience than a device that requires the user to initiate bites. Herlant [39] designed an HMM to predict bite timing in dyadic robot-assisted feeding. However, her model only considered the social cues of the user. Bhattacharjee et al. [40] found users preferred less intrusive interfaces in a social dining scenario, specifically a web interface over a voice interface. Our work aims to build non-intrusive bite timing strategies by focusing on learning when to feed a user in triadic scenarios while using implicit social features from all diners.

Particularly, bite timing is important because the consequences of presenting a bite to the diner earlier than expected is poorly tolerated. This can include an interruption to conversation or to finishing chewing the prior bite. The consequences of presenting a bite later than desired can include frustration towards the robot and disruption of the natural flow of conversation during the meal. Parallels can be drawn to interruptibility research on finding the most appropriate timing to probe a user. Researchers have found that people performed best on a task if interruptions were mediated rather than timed immediately or on scheduled intervals [41, 42], often mediated based on modeling contextual and social factors [43–46].

A socially-aware robot-assisted feeding system should be designed such that if needed, the user should be able to communicate these intentions via multiple different modalities such as body language, gaze, or speech. These various modalities have been found to be effective in modeling social interactions [47–50]. Capturing these natural social interactions in computational models are likely crucial to provide accurate bite timing without distracting users from the social ambiance.

## 3    Problem Formulation

The objective of the bite timing prediction problem in robot-assisted feeding with a single diner is to predict the timing of *when* this user will take a bite of food by capturing their signals $\mathbf{U}$ such as voice, body gestures, head movements or speaking status. We define the proper timing for when a robot should feed as when the user *intends* to take a bite of food. It takes input signals $\mathbf{U}(t_0 : t)$ from time $t_0$ to time $t$ and learns a function $\mathcal{F}(\mathbf{U})$ to predict a Boolean $y(t + h) = \mathcal{F}(\mathbf{U}(t_0 : t))$, which indicates whether the user intends to take a bite in the time horizon $h$ and trigger a bite transfer at time $t + 1$. When a person lifts their fork off the plate to eat, they intend to take a bite of food, where this time horizon $h$ is the time it takes to transfer the food to their mouth from their plate.

In this paper, we consider a social variant of the bite timing prediction problem where a user is interacting with two co-diners. Our goal is to predict the timing of a user to take a bite of food based on the social cues within the interaction. From an initial time $t_0$ to time $t$, the user receives social signals $\mathbf{L}(t_0 : t)$ and $\mathbf{R}(t_0 : t)$ from their left and right conversational co-diners, respectively. Given these external social signals and the target user's own history of signals $\mathbf{U}(t_0 : t)$, we aim to predict $y$. We note that it may not always be possible to track the same set of features for a user and their co-diners. Therefore, for some time range $k = t - t_0$ and feature dimensions $n, m$ for the user and co-diners respectively, $\mathbf{U} \in \mathbb{R}^{k \times n}$ while $\mathbf{L}, \mathbf{R} \in \mathbb{R}^{k \times m}$, where $n$ does not necessarily equal $m$. The function to learn is:

$$y(t + 1) = \mathcal{F}(\mathbf{U}(t_0 : t), \mathbf{L}(t_0 : t), \mathbf{R}(t_0 : t))$$

## 4    Model: SOcial Nibbling NETwork (SoNNET)

We present the SOcial Nibbling NETwork (SoNNET) that predicts when a user has the intention to eat based on various social signals. We selected features to represent both human eating and social

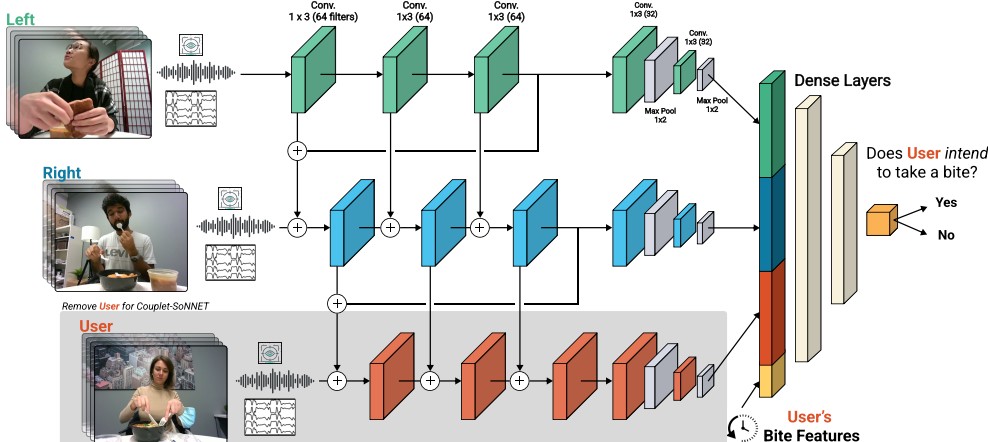

Figure 2: **Triplet-SoNNET** contains three interacting channels for features of a **target user** and two co-diners. Each channel concatenates the input of time, gaze, speaking and skeleton features from each single diner. **Couplet-SoNNET** eliminates all features from the **target user** by dropping the last channel; however, it continues to use the user's bite features. Batch normalization layers are not shown in the figure.

behavior: bite features, which include the number of bites taken so far and the time since the last bite of food $b \in \mathbb{R}^2$, a diner's gaze and head pose direction $d \in \mathbb{R}^4$, binary speaking status $s \in \{0, 1\}$, and face and body keypoints $o \in \mathbb{R}^{168}$ from OpenPose [51]. We note that, in our case, the bite features $b$ are computed only for the user and not the co-diners, since we do not estimate in real-time whether a co-diner is taking a bite of food. Thus, for a time interval $k = t - t_0$, these features are temporally stacked to construct the input signals $\mathbf{U} \in \mathbb{R}^{k \times 175}, \mathbf{L} \in \mathbb{R}^{k \times 173}, \mathbf{R} \in \mathbb{R}^{k \times 173}$ for the user, left co-diner, and right co-diner, respectively.

Recently, convolutional neural networks (CNNs) have demonstrated significant success for multi-channel time series classification from various kinds of signals [52–54]. Wu et al. [46] proposed PazNet: a multi-channel deep convolutional neural network which is able to handle inputs of different dimensions. PazNet is designed to predict the interruptibility of individual drivers. However, the information of different channels is not shared, and it lacks ability to capture social interactions among multiple people.

We design the Social Nibbling NETwork (SoNNET), a new model architecture which follows a multi-channel pattern allowing multiple interconnected branches to interleave and fuse at different stages. We create input processing channels for each diner, then add interleaving tunnels between each convolutional module and adjacent branches. The information capturing visually-observable behaviors between the diners is allowed to flow between the frames and channels. We conjecture that our model will learn a socially-coherent structure, allowing the model to implicitly represent the diners in an embedding space. Therefore, each channel has the same structure but does not share the same weight parameters. To help capture informative features, we performed dimension-reduction after the interleaving components using max pooling layers and $1 \times 1$ convolutional layers. These per-diner channels are concatenated and then followed by two dense layers for classification, which decides whether the user intends to feed or not. For SoNNET, the range between $t$ and $t_0$ is six seconds. The social signals in this range are used to predict a user's bite intentions.

**Triplet-SoNNET.** For modeling the bite-timing prediction of three users with no mobility limitations, we propose Triplet-SoNNET which uses social signals from the left and right co-diners $\mathbf{L}, \mathbf{R}$ and signals from the user $\mathbf{U}$. Depicted in Fig. 2, Triplet-SoNNET ensures that the features from other co-diners $\mathbf{L}, \mathbf{R}$ interleave into the target user's features $\mathbf{U}$.

**Couplet-SoNNET.** To run Triplet-SoNNET in a robot-assisted feeding setting, there would be a distribution shift in the kinds of signals a target user outputs. Our goal is to feed people with mobility limitations while they are engaged in social conversations. The features from someone self-feeding are inherently different from someone using a robot-assisted feeding system. In the case of body pose, a target user with C3-C5 SCI would be largely still, which is different from the training data. Our Human-Human Commensality Dataset consists of adult diners with no mobility limitations, thus applying a trained Triplet-SoNNET model to robot-assisted feeding of a user with mobility limitations would be out-of-distribution. Although our target users with C3-C5 SCI cannot

move their arms to feed themselves, there is still a large spectrum of severity in mobility limitations depending on the users' conditions. From our discussions with key stakeholders, caregivers, and occupational therapists we design, Couplet-SoNNET, where we ignore most social signals from the target user by removing the last channel in Triplet-SoNNET. Therefore, the intention to feed $y(t + 1) = \mathcal{F}(\mathbf{U}_b(t_0 : t), \mathbf{L}(t_0 : t), \mathbf{R}(t_0 : t))$, where $\mathbf{U}_b \in \mathbb{R}^{k \times 2}$ are the user's bite features for $k = t - t_0$. The user's bite features, such as the time since the last bite and the number of bites since the onset of the feeding activity, are the only social signals from the target user. Additional discussion on this design choice can be found in App. 8.2.2.

## 5 Human-Human Commensality Dataset (HHCD)

We introduce a novel Human-Human Commensality Dataset (HHCD) of three participants with no mobility limitations eating in a social scenario. We used this dataset to develop models that predict a diner's intention to take a bite of food while taking into account subtle social cues. We deployed the trained models in a social robot-assisted feeding setting where one diner is fed by a robot. Beyond predicting bite timing, we are excited for the robot learning community to find other interesting challenges within our dataset that leverage understanding social dynamics.

**Data Collection Setup.** We recruited 90 people among our Institution-affiliated fully-vaccinated students, faculty, and staff to eat a meal in a triadic dining scenario. Each participant was 18+ years old and took part in the study only once. The study setup is illustrated in Fig. 1 (left). There are three cameras (mutually at $120°$) in the middle of the table, each capturing one participant, and a fourth camera capturing the whole scene. All four cameras are Intel RealSense Depth Cameras D455 [55]. The scene audio was captured by a ReSpeaker Mic Array v2.0 [56] placed in the middle of the table. The ReSpeaker microphone array has four microphones arranged at the corners of a square and estimates the direction of sound arrival. For the study setup measures, see App. 8.1.2.

Participants were free to bring any kind of food and any utensil with them. They could also bring a drink (some drank from a cup, others from a bottle or both, with or without a straw) and were provided with napkins. Before the study, each participant was asked to fill in a pre-study questionnaire about their demographic background, relationship to other participants, and social dining habits. The experimenter then asked them to eat their meals and have natural conversations. At this point, the experimenter started the recording and left the room. When *all* three participants finished eating or after 60 minutes have passed, whichever was earlier, the experimenter stopped the recording and asked participants to fill in a post-study questionnaire about their dining experience. The specific questions asked in both pre/post-study questionnaires can be found in App. 8.1.3. The study was approved by Cornell's IRB.

**Data Annotation.** We annotated each participant's video based on their interactions with food, drink, and napkins. In particular, we annotated *food_entered*, *food_lifted*, *food_to_mouth*, *drink_entered*, *drink_lifted*, *drink_to_mouth*, *napkin_entered*, *napkin_lifted*, *napkin_to_mouth*, and *mouth_open* events. We chose these events as they are key transition points during feeding. We spent 151 hours annotating and used the ELAN annotation tool [57]. We assigned the annotation value $\in \{fork, knife, spoon, chopsticks, hand\}$ based on the utensil performing the food-to-mouth handover. While annotating, we also noted down per-participant food types and observations of interesting behaviors. All annotation types with detailed rules are provided in App. 8.1.4.

**Data Statistics.** There were 56 female and 34 male participants, and their ages ranged 18-38 ($\mu = 22$, $\sigma = 3$) years. Session durations ranged 21-55 ($\mu = 37$, $\sigma = 9$) minutes and 1 session was at breakfast, 10 at lunch, and 19 at dinner time. For additional dataset statistics, see App. 8.1.5.

For a summary of all available data in the dataset and its detailed analysis, see App. 8.1. For the purposes of this work, we only consider bite features, speaking status, gaze and head pose, and body and face keypoints.

## 6 Model Evaluation on Human-Human Commensality Dataset

In this section, we evaluate Triplet- and Couplet-SoNNET against other models on the HHCD. In particular, we compare against a regularized linear SVM trained with SGD to evaluate performance of linear classifiers. We also consider a Temporal Convolution Network (TCN) [58, 59], which uses causal convolutions and dilations to represent temporal data. TCNs have been found to perform better than LSTMs and GRUs on temporal anomaly detection [60] and robot food manipulation tasks [20], therefore they would provide a strong baseline to compare our models to. We also perform an

Table 1: Ablation study on different modalities from various data sources. We use average over LOSO cross-validation.

| Method | Acc. | Prec. | Rec. | F1 | nMCC |
|---|---|---|---|---|---|
| Triplet-SoNNET | **0.820** | 0.861 | 0.871 | 0.862 | **0.772** |
| - Speaking Status | 0.816 | **0.864** | 0.863 | 0.856 | 0.771 |
| - Gaze & Head Pose | 0.815 | 0.863 | 0.863 | 0.856 | 0.769 |
| - Bite Features | 0.781 | 0.832 | 0.855 | 0.834 | 0.727 |
| - Body & Face | 0.820 | 0.854 | **0.886** | **0.865** | 0.771 |

Table 2: Model comparison on LOSO cross-validation over 29 sessions.

| Method | Acc. | Prec. | Rec. | F1 | nMCC |
|---|---|---|---|---|---|
| Always Feed | 0.72 | 0.72 | 1 | 0.83 | 0.5 |
| Linear SVM (SGD) | 0.68 | 0.82 | 0.77 | 0.74 | 0.64 |
| Triplet-TCN | 0.82 | 0.82 | **0.96** | **0.88** | 0.72 |
| Triplet-SoNNET | **0.82** | **0.86** | 0.87 | 0.86 | **0.77** |
| Couplet-TCN | 0.73 | 0.73 | **0.98** | 0.83 | 0.55 |
| Couplet-SoNNET | **0.76** | **0.78** | 0.96 | **0.85** | **0.66** |

ablation study to investigate the importance of various modalities. Implementation details about baseline models, SoNNET, and training procedure can be found in App. 8.2.

For training, we use 6811 *food_lifted* annotations as positive training labels since they precede an actual bite of food and indicate an intention to eat. We use a time interval of $k = t - t_0 = 6$ seconds because it takes roughly 6 seconds for the robot to move from its wait position to feeding the user. Since bite actions are sparsely distributed over time, we select 2486 6-second clips as negative samples that are in the middle of two *food_lifted* annotations. All reported models are trained with leave-one-session-out (LOSO) cross-validation to evaluate generalizability to new groups of people. Due to an issue with recording, we train over 29 sessions if speaking status features are used.

The user's bite features $b \in \mathbb{R}^2$ (time since last bite and the number of bites eaten since the start) are indicators of eating rate. To ensure this feature is not dominated by higher dimensional features, we scale the size of the input by $\gamma$. This hyperparameter $\gamma$ scales $b \in \mathbb{R}^2 \to b \in \mathbb{R}^{2\gamma}$. We selected $\gamma = 100$ after a grid search over the training set on the TCN and SoNNET models.

**Evaluation Metrics.** A high recall indicates that our model can reliably feed when it should. In contrast, a high precision indicates that a model tends to be stricter in deciding when to feed. Due to our dataset imbalance, the average accuracy across 29 sessions for a model that predicts it should always feed is 71.56%. This classifier achieves perfect recall, and relatively high precision, causing the model to have a high F1 score. It is clear that given this class imbalance, a high F1 score poorly represents the capabilities of this model. To evaluate our model effectively, we consider the normalized Matthews Correlation Coefficient (nMCC) in addition to F1 score, precision, recall, and accuracy. Unlike F1 score, nMCC considers the size of the majority and minority classes, and can only produce high scores if a classifier is able to make correct predictions for a majority of both the negative and positive classes [61]. A value of 0.5 indicates random prediction, while 0 is inverse prediction and 1 is perfect prediction.

**Effects of Modality.** We are interested in investigating features that are the most informative for designing a good bite timing predictor in social dining. We perform a feature ablation study on the Triplet-SoNNET model, as shown in Table 1. We selectively remove feature streams, such as body and face data from OpenPose, gaze and head features from RT-GENE, speaking status signals, and the user's bite features. We find that users' bite features such as the time since last bite and the number of bites are important, as accuracy drops drastically without them. Intuitively, we believe this feature is important because a user's bite features are a proxy for their level of eating rate. We also see that without body and face features, F1 and recall slightly increase. This could be due to the fact these data streams are noisy; however, as indicated by the lower accuracy and nMCC when removing OpenPose features, these features are useful.

**Effects of Model Type.** Table 2 shows the outcomes of various model comparisons when trained using LOSO. We compare performance of Triplet-SoNNET against a linear SVM and TCN trained on all three diners. We call this TCN a Triplet-TCN. Triplet-TCN has all the diners' features concatenated per-timestep, and we compare this result to Triplet-SoNNET. We find that Triplet-SoNNet achieves higher accuracy and nMCC compared to all other models; however, it performs worse on recall and F1 score compared to Triplet-TCN. In our scenario, we want to ensure that the robot feeds when it should and does not feed when it should not. A bite prediction model that overfeeds or underfeeds is not ideal. A high nMCC balances the roles of recall and precision and better represents whether a classifier should or should not feed. Therefore, for our scenarios, Triplet-SoNNET is a more effective predictor of bite timing than other models trained on all three diners.

**Effects of Social Scenario.** We are interested in comparing the ability of models to learn social behaviors using only two co-diners' features rather than having full observability. We compare

Couplet-SoNNET to a similarly-named Couplet-TCN trained on two co-diners' features and a user's bite features. As expected, Couplet-TCN and Couplet-SoNNET perform worse than their Triplet-counterparts, with Couplet-TCN being close to random prediction with an nMCC of $0.5539$ while Couplet-SoNNET has an nMCC of $0.6648$. We find that Couplet-SoNNET performs better than Couplet-TCN. This result reveals Couplet-SoNNET is able to understand social signals better than a predictor that always feeds. This implies that it is possible to predict the behavior of a user using only their co-diner information, which indicates that there is social coordination in human-human commensality. These findings also suggest that social signals were captured by the interleaving structure of the SoNNET models.

## 7 Transferring from Human-Human to Human-Robot Commensality

Our objective is to develop a bite timing strategy for a robot that feeds a user in a social dining setting. We design a study where users evaluate the effect of different bite timing strategies on their overall social dining experience. To simulate robotic caregiving scenarios for people with upper-extremity mobility limitations, we instructed users to not move their upper body. This study was approved by Cornell's IRB.

**Experimental Setup.** We evaluate a learned bite timing strategy against two baseline bite timing strategies inspired by our conversations with care recipients, occupational therapists, and caregivers who told us how they know when to feed. The strategies are further depicted in Fig. 3:

1. **Learned Timing.** This social, fully autonomous bite timing strategy feeds based on our Couplet-SoNNET model's output. We sample this model every three seconds with the last six seconds of preprocessed features at a rate of 15 frames per second. This approach takes into account the social context. Since we want to evaluate the generalization performance, we train Couplet-SoNNET on 80% of the HHCD data and use the remaining 20% of HHCD data to select early-stopping criteria.
2. **Fixed-Interval Timing.** This fully autonomous bite timing strategy feeds every $44.5$ seconds, which is a scaled average time a robot should take to feed a user after it has picked up a food item. To derive this value, we first find the appropriate scaling factor between human motion from the HHCD and robot motion. We note the average time for a human from the *food_entered* transition to *food_lifted* transition is $9.9$ seconds. The robot end-effector motion is not designed to match the human speed but rather to be perceived as safe and comfortable to a user being fed. We find the equivalent key transitions for the robot to be $5x$ slower than a human. Since we define the intention to take a bite as when the food is lifted, the robot should take $49.5$ seconds to feed a user after picking up a food item. Given the robot takes roughly $5$ seconds to move to its wait position after picking the food, the robot waits $44.5$ seconds. Further details about this wait-time can be found in App. 8.3.2.
3. **Mouth-Open Timing.** This partially autonomous bite timing strategy feeds only when the user prompts the robot by opening their mouth. The target user is prompted each time by the robot saying "When ready, look at me and open your mouth". This approach gives the user explicit control of when the robot should feed [40].

The robot user is seated on a wheelchair mounted with a Kinova Gen3 6-DoF arm [62], which is used to feed the participant (Fig. 3, left). For discussion of our bite timing strategies, the use of voice prompting, and implementation details of the robot study, see App. 8.3.2-8.3.4.

**Experimental Procedure.** In this study, participants are seated in a similar setup as that used for HHCD data collection in Sec. 5. All participants were asked to bring their own food, and each group chose who would be fed by the robot. We recruited 30 participants over 10 sessions. There were 16 female and 14 male participants, and their ages ranged from 19-70 ($\mu = 27$, $\sigma = 9$) years.

A single trial consists of bite acquisition, followed by one of the three bite timing strategies, then bite transfer. For bite acquisition, the robot alternates feeding the user cantaloupes and strawberries. We chose these fruits due to their high acquisition success rates [24]. We used the bite acquisition strategies and bite transfer strategies from [25, 26]. All participants take a survey after each trial, which administers a forced-choice question on the participants' preferences between the previous and current conditions. Each pair of comparisons between any two conditions occurs three times, leading to ten trials. The condition orderings are counterbalanced over ten trials. Additionally, we ask participants whether they felt the robot fed them too early, on-time, or too late. The experiment questionnaire after each trial further includes questions about bite timing appropriateness, distractions due to the robot, ability to have natural conversations, ability to feel comfortable around the

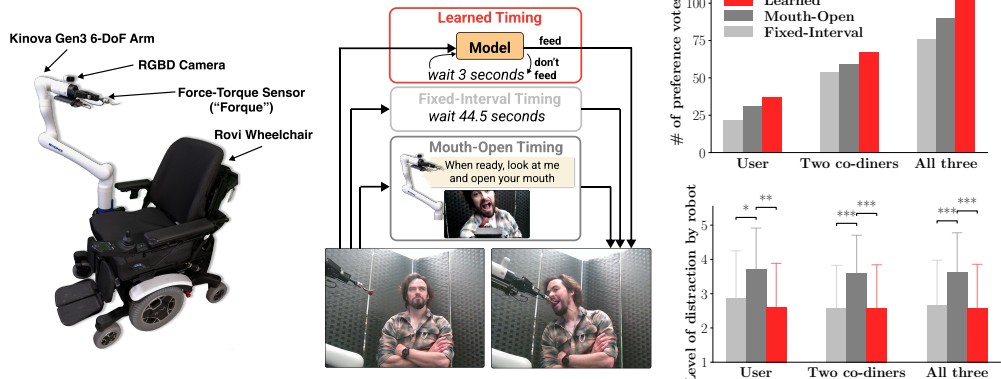

Figure 3: **Left:** We use a 6-DoF Kinova Gen3 robotic arm [62] on Rovi wheelchair [63]. **Middle:** User study conditions/bite timing strategies: Learned, Fixed-Interval, and Mouth-Open Timings. **Top right:** Preferences for bite timing strategies rated by users, two co-diners, and all three diners. **Bottom right:** Level of distraction by the robot perceived by users, two co-diners, and all three diners on a Likert scale 1-5 (agreement with "I felt distracted by the robot"), for each bite timing strategy. $*, **, ***$ denote statistically significant differences with $p_{0.05}, p_{0.005}, p_{0.0005}$ respectively.

robot, as well as system reliability and trust in the robot [64]. For details on user study questionnaires see App. 8.3.6. To avoid interruptions in social conversations due to the presence of a robot in human groups, we provide the participants with a list of questions (see App. 8.3.5), which they could optionally use to help get the conversation started at each trial, similarly to previous work [39].

**Results and Discussion.** As shown in Fig. 3 (top right), users and co-diners preferred the Learned strategy for bite timing as compared to Fixed-Interval or Mouth-Open Timing. This confirms that our insight to incorporate social signals in model structure (SoNNET) improves bite timing prediction. These results using Couplet-SoNNET also imply that it is possible to predict the behavior of a user using only their co-diner information, which indicates that there is social coordination in human groups even in the presence of a robot. In Fig. 3 (bottom right) we further compared the level of distraction by the robot as perceived by participants. We performed Kruskal-Wallis H-tests and Tukey HSD post-hoc tests and found that Mouth-Open Timing distracts dining participants significantly more than Learned or Fixed-Interval Timing. We believe this is because the Mouth-Open strategy prompts the user using a voice interface, which can disrupt the rhythm of conversation. Even though the participants had a clear preference for the Learned strategy when given a forced-choice, when asked to individually rate the conditions using a 5-point Likert scale, interestingly we could not find any statistically significant differences between Mouth-Open Timing and Learned Timing. This is probably because the Mouth-Open Timing strategy provides full control of bite timing to the users themselves. Note, regardless of the conditions, the users found the system comfortable, reliable, and trustworthy. Detailed analysis is given in App. 8.3.7.

**Limitations.** There is a risk that our results from human-robot user studies on adults with no mobility limitations may not generalize to those with people with mobility limitations. People with mobility limitations may have different preferences and cognitive workload associated with a robotic intervention. Although our target diner is not a person with such C3-C5 SCI, our model does not use their movements to infer when to feed. As transferability is a function of their behavior, our experiments demonstrate good transferability across scenarios. We expect it to similarly transfer to users with C3-C5 SCI, though it remains to be investigated in future work. We also made multiple assumptions when transferring our results from human-human to human-robot commensality scenarios. During human-human commensality, the user was self-feeding whereas in human-robot commensality the user was being fed. We also assumed that the addition of a robot into a human-human commensality scenario does not change the social dynamics of the diners significantly. Given these assumptions, it would be interesting to see how our models perform when trained on similar human-robot commensality scenarios. Finally, it is an open question as to how these models would perform with groups of different cultures. Social science literature on commensality studied the interplay between factors such as culture [65, 66], age [67], and social context [68] on how long eating takes and what people are eating. We are excited about the potential to study how the presence of a robot can alter the communal act of eating together across cultures. This motivates further investigation into human-robot commensality, both from technical and societal perspectives.

**Acknowledgments**

The authors would like to thank Rajat Jenamani, Rishabh Madan, and Sidharth Vasudev for their help with setting up and running the robotics user study. This work was funded in part by the National Science Foundation IIS (#2132846). This work was also in part sponsored by the Office of Naval Research (N00014-19-1-2299). Any opinions, findings, and conclusions or recommendations expressed in this material are those of the author(s) and do not necessarily reflect the views of the Office of Naval Research.

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
