# OpenReview forum: "Human-Robot Commensality: Bite Timing Prediction for Robot-Assisted Feeding in Groups"
_robot-learning.org/CoRL/2022/Conference — CoRL 2022 Poster_

### Official Review · Reviewer_iJwW · 2022-07-27

**Originality:** Very Good
**Technical Quality:** Good
**Clarity Of Presentation:** Very Good
**Impact:** 3

**Recommendation:**

Weak Accept: I recommend accepting the paper, but will not argue for my recommendation if the majority of other reviewers have a different opinion.

**Summary:**

The paper provides a multi-modal human-human commensality Dataset (HHCD) containing 30 groups of triads eating together, with rich annotation of keyframes of activities (e.g. picking food, taking bites, etc). Based on this, the authors present a data-driven model to predict when a robot should feed a user during social dining scenarios based on bite history and features of only co-diners (e.g. facial features, gaze, etc). Their user studies show that their model is more natural and less interruptive than other baselines.

**Issues:**

Here are a couple of concrete suggestions that might help further improve the paper.
- If time allows, re-evaluate the Mouth-Open Timing method with only one-time prompting at the beginning.
- Clarify why Fixed-Interval Timing needs a 5x scaling factor
- Clarify what modalities the model relies on when making predictions. The model is clearly learning something meaningful (otherwise nMCC would be 0.5), so it’s a bit confusing why the nMCC only drops by a little in most ablations in Table 1. In the paper, it’s mentioned that Body & Face feature from OpenPose is important, but it’s unclear that is the case from Table 1.


**Quality Of The Limitations Section:**

Limitations are addressed clearly

**Reviewer Expertise:**

4: The reviewer is confident but not absolutely certain that the evaluation is correct

**Robotics Focus:**

Sufficient demonstration on hardware

**Strengths And Weaknesses:**

This paper is very well-motivated and investigates a very underexplored research area of robot assistive feeding in social scenarios. The problem domain is focused and well-defined: predict the right timing to feed the human. The collected dataset HHCD is relatively large-scale and contains diverse demographics and social cues, so it seems to be generally useful for people to further investigate the topic of commensality. The details of the data collection and annotation are well-documented and explained in the paper and in the appendix. On the model side, the authors proposed an interesting interleaving architecture to fuse features from co-diners; their method is also able to consistently beat a strong baseline (TCN) in different scenarios. The most interesting finding is that their model shows evidence that it’s possible to predict the behavior of a user using only their co-diner information, suggesting some level of social coordination. The paper overall is clearly written and easy to follow.

However, the paper is not without shortcomings. In section 7, which is the ultimate test of deploying the model on a human-robot commensality scenario, the two alternative bite timing strategies raise some questions. For the fixed-interval timing, it’s unclear what the authors mean by “the equivalent key transitions for the robot to be 5x slower than a human” and why scaling the time by 5x is ideal. If the average time between food_lifted and food_entered is around 10 seconds, presumably the target user wants the robot arm to do the same, rather than waiting for 40+ seconds? More importantly, for mouth-open timing, it seems unnecessary for the target user to be prompted each time by the robot via the voice interface. It seems more natural for the user to be prompted only once at the beginning of the trial so that the user understands how things work. Going forward for every bite, the user should only need to look at the robot and open their mouth. As a result, it’s not fully convincing that the learned method is necessarily preferred over the other two baselines. Another interesting result that raises questions is the ablation study in Table 1. It seems that the model is mostly relying on the bite features from the target user alone to make predictions. It’s a bit unclear whether the gaze, head pose, body, and face features of the diners actually help the model.


**Summary Of Recommendation:**

Although the paper tackles a very interesting, underexplored domain and did a thorough job of collecting a large-scale dataset and conducting human subject experiments, the main result in Section 7 isn’t fully convincing. Studying and understanding robot-assisted commensality is an extremely important topic, but it’s still unclear whether predicting bite timing is the key and whether a learned method necessarily is preferred over a much simpler heuristics-based method or a user-triggered mechanism. However, I am happy to hear the authors’ responses and revise my rating/recommendation.

---

> ### Author Response · Authors · 2022-08-25
> **Individual Response to Reviewer iJwW (1/2)**
>
> We thank the reviewer for a very nice and concise summary of our paper’s strengths. We also want to specifically point out one of the strengths that was pointed out below:
>
> **R-iJwW-S >** _The most interesting finding is that their model shows evidence that it’s possible to predict the behavior of a user using only their co-diner information, suggesting some level of social coordination._
>
> We were excited to find this result in our human-human commensality data. We are inspired to leverage this fact for new challenges within human-robot commensality, and we hope this insight can help spawn new research on robot learning in groups.
>
> **R-iJwW-1 >** _For the fixed-interval timing, it’s unclear what the authors mean by “the equivalent key transitions for the robot to be 5x slower than a human” and why scaling the time by 5x is ideal. If the average time between food_lifted and food_entered is around 10 seconds, presumably the target user wants the robot arm to do the same, rather than waiting for 40+ seconds?_
>
> We thank the reviewer for noting the 5x scaling factor. Our goal was to find a simple fixed-time heuristic that would be a strong and fair baseline. We used data from our human-human commensality dataset (HHCD) to decide this wait time. From HHCD, we find that a human on average takes 1.8s from lifting a food item off a plate / bowl (food_lifted) to bringing it to the mouth (food_to_mouth). The robot’s equivalent approach duration is on average around 9 seconds (taking into account the variable motion planning time). Though the robot could mechanically move at a faster speed, we chose the speed that would feel safe and comfortable to a user when they are being approached (fed) by a robot with a fork. We determined this velocity of our robot to be perceived as safe and comfortable based on [1] which explicitly did a study on what approach speeds are preferred by people with mobility limitations. This scaling factor (9s / 1.8s = 5) between robot speeds and human speeds is thus user-inspired. Once we determined this scaling factor, we use this same scaling factor to scale up the bite timing from HHCD (9.9s) to human-robot commensality (9.9s*5 - 5s [for robot bite acquisition to bite-timing waiting position] = 44.5s) to make sure that the proportion of time for different phases of feeding (bite acquisition - bite timing - bite transfer) are all proportional and balanced. We would also like to note that although the average time for “food_entered → food_lifted” was 9.9s in HHCD, the standard deviation was 27.3s. So a wait-time of 44.5s is roughly 1 standard deviation away from the equivalent annotation in the HHCD data. Our learned model, on the other hand, used social cues from the user and the other co-diners to determine the bite-timing that results in variable bite-timing during the course of the meal, which from the real robot user study, is preferred by the participants. We will revise our text in the paper to make this clearer.
>
> [1] T. Bhattacharjee, E. K. Gordon, R. Scalise, M. E. Cabrera, A. Caspi, M. Cakmak, and S. S. Srinivasa. Is more autonomy always better? exploring preferences of users with mobility impairments in robot-assisted feeding. In 2020 15th ACM/IEEE International Conference on Human-Robot Interaction (HRI). IEEE, 2020
>
> **R-iJwW-2 >** _More importantly, for mouth-open timing, it seems unnecessary for the target user to be prompted each time by the robot via the voice interface. It seems more natural for the user to be prompted only once at the beginning of the trial so that the user understands how things work. Going forward for every bite, the user should only need to look at the robot and open their mouth. As a result, it’s not fully convincing that the learned method is necessarily preferred over the other two baselines_
>
> Thank you for asking this question. Our study is a counterbalanced study design of two fully autonomous conditions (learned and fixed-interval) and one partially autonomous condition (mouth-open timing). The users are not seeing the mouth-open condition sequentially. It is always followed and preceded by a different condition. The prompting of the voice interface indicates that user input will be required for the current trial, which the other two conditions do not require. The users need to know whether the feeding is triggered by opening their mouth or whether it is triggered automatically. If the study was not constructed as such, then we would be unable to include a forced-choice preference question. These pairwise forced comparisons allowed us to discover which method was most preferred.
>
> We designed this mouth-open condition by going to assisted care living facilities and discussing with care recipients, occupational therapists, and caregivers as to how they know when to feed. Caregivers mentioned that they estimate bite timing when care recipients open their mouth and look at them. We believe our baselines are fair and inspired by real-world usage.

---

> > ### Comment · Reviewer_iJwW · 2022-08-25
> > **Discussion with Authors**
> >
> > Thank you for your detailed response
> >
> > **R-iJwW-1**
> >
> > Thank you for clarifying the 5x scaling factor in detail.
> >
> > **R-iJwW-2**
> >
> > I am still not sure I am fully convinced and I would love to discuss more with the authors. First of all, I fully agree that the mouth-open condition is inspired by real-world usage and is very natural. I am just not sure if voice prompting is needed. From the author's response, I understand that voice prompting is needed because the authors interleave the three conditions during their experiments (A -> B -> C -> A..., or something like this?).
> >
> > > If the study was not constructed as such, then we would be unable to include a forced-choice preference question. These pairwise forced comparisons allowed us to discover which method was most preferred.
> >
> > I am not fully convinced that the study has to be conducted in this way, but I am open to hearing more responses from the authors. What if the user experiences 5 mins of condition A, then 5 mins of condition B, and then 5 mins of condition C, and then rates them in the end? The exact condition order can be randomly shuffled. I think getting rid of the voice prompting will make the mouth-open condition closer to real-world applications. Intuitively, voice prompting is the main culprit of creating social distractions, not the "mouth-opening" or "looking-at-the-camera" part.
> >
> > **R-iJwW-3**
> >
> > Got it. Thanks!

---

> > > ### Author Response · Authors · 2022-08-26
> > > **Discussion with Reviewer iJwW (1/2)**
> > >
> > > > What if the user experiences 5 mins of condition A, then 5 mins of condition B, and then 5 mins of condition C, and then rates them in the end? The exact condition order can be randomly shuffled.
> > >
> > > We thank the reviewer for bringing up their proposed study design and for willing to discuss this with us. We had initially considered a similar experimental design, where we feed 2 or 3 times in each condition that follows sequentially. Note, bites can be of variable time length in each condition, so the timing would not be the same “x” minutes for each of those conditions. Instead of the sequential method, we chose our counterbalanced forced-choice study design because it can mitigate “recency biases” [1] in the survey questions. Let us explain our choices in detail below:
> > >
> > > The reviewer’s proposed study design (which matches our initially considered study design) proposes a fixed-number of bites for the three conditions, which are randomly shuffled. Then the participant rates them in the end. Let us say we do the experiment as A→B→C. Rather than a pairwise-forced choice question, the participant would rate their most preferred choice at the end. This approach would suffer from the recency tendency, where the participants will likely struggle to recall how they felt when they ate 9 bites ago (for 3 bites per condition). In regards to the Likert questions, there would be more biases from order effects in the questionnaire.
> > >
> > > Our chosen experiment design is a within-subjects repeated-measures design where the conditions are counterbalanced such that A→B and B→A occur a total of 3 times and there is only 1 bite per condition at one time. This helps mitigate the recency tendency and guarantees that within each session, there is a tie-breaker. Within one session, we thus get 9 comparisons from 10 trials. Across 10 sessions we ensure that each ordered pair occurs an equal number of times. This gives us a total of 30 comparisons, 15 A→B and 15 B→A (and similarly for BC, AC). Beyond forced-choice questions, we get less-biased information in the Likert questions from 10 counterbalanced trials. Note that, forced-choice questions are also not completely without biases towards the most recently-observed condition or absolute judgments [2]. However, this study-design is generally better for eliciting preference data [3] and has less “recency bias” [1] than study-design conditions that are presented sequentially with preference-based questions presented at the end (“x” mins of condition A, followed by “x” mins of condition B, and then “x” mins of condition C and then rating in the end). Moreover, by additionally counterbalancing our forced-choice design (A→B and B→A), we further mitigate possibilities of any remaining recency bias.
> > >
> > > We would also like to take this opportunity to mention that we had also considered a third study design that has a fixed-number of bites per condition but uses our study design of counterbalancing condition orders. Let us say we do an experiment as A→B→C→B→A→C. If we have 2 bites per condition, it would take 12 bites to complete the study. We get 5 comparisons (2 AB, 2 BC, 1 AC) with no tie-breaker. With a slightly higher experiment length, we get half of the preference information. But again for this study-design, there would be more biases from order effects in the questionnaire. Although counterbalancing A→B and B→A does help reduce the recency tendency, having 2 bites per condition reintroduces recency biases since they have to recall what they felt over two bites ago. If we increased the number of bites per condition, it would exacerbate this issue. Therefore, we believe our current study design to be the most appropriate to elicit the preferences of users. We will revise the text of the paper to clarify this, and add additional explanations on our choices in the appendix.
> > >
> > > [1] M. Mehrani and C. Peterson. Recency Tendency: Responses to Forced-Choice Questions. Applied Cognitive Psychology. 2015
> > >
> > > [2] J. Starns, T. Chen, A. Staub. Eye movements in forced-choice recognition: Absolute judgments can preclude relative judgments. Journal of Memory and Language. 2017.
> > >
> > > [3] S. Sankaran, J. Derechin, N.A. Christakis. CurmElo: The theory and practice of a forced choice approach to producing preference rankings. PLoS One. 2021

---

> > > > ### Comment · Reviewer_iJwW · 2022-08-26
> > > > **Further Discussion with Authors**
> > > >
> > > > Thank you very much for your detailed explanation. I appreciate the authors' efforts to eliminate biases as much as possible in their experiments. I must admit that the authors probably have more expertise in designing human experiments than I do, so the AC should feel free to take my comments with a grain of salt.
> > > >
> > > > However, I still believe the voice prompting at every bite puts this baseline at an unfair disadvantage. Let's take a step back and think about how we would deploy the user-trigged baseline in the real world. In a real-world scenario, the user probably doesn't need voice prompting: it's sufficient for them to just look at the camera and open their mouthes if they want the next bite. If possible, I would still like to see some new experimental results while voice prompting is removed. It would be great to show the new results align with the original results. I fully agree that there will be recency biases, and that's why I suggest randomly shuffling A, B, C (e.g. 5 bites of A, 5 bites of B, 5 bites of C), which should "average out" the recency biases if there is decent population size.
> > > >
> > > > Finally, I fully agree that investigating different types of user-triggered interfaces is outside the scope of the paper. I didn't intend to ask the authors to do so in my original comment. What I tried to convey is that I believe the user-triggered baseline should not require any disruptive interface (voice, silent light-based, or web-based) at all. The users should be able to just look at the camera and open their mouths, without any form of prompting.
> > > >
> > > > Happy to discuss this further! Thanks!

---

> > > > > ### Author Response · Authors · 2022-08-27
> > > > > **Further Discussion with Reviewer iJwW (1/2)**
> > > > >
> > > > > > "However, I still believe the voice prompting at every bite puts this baseline at an unfair disadvantage. Let’s take a step back and think about how we would deploy the user-triggered baseline in the real world. In a real-world scenario, the user probably doesn’t need voice prompting: it’s sufficient for them to just look at the camera and open their mouth if they want the next bite."
> > > > >
> > > > > > "What I tried to convey is that I believe the user-triggered baseline should not require any disruptive interface (voice, silent light-based, or web-based) at all. The users should be able to just look at the camera and open their mouths, without any form of prompting."
> > > > >
> > > > > We thank the reviewer for continuing this discussion with us. From our initial pilot studies, we found that the user’s attention was usually focused on the conversation with the co-diners, not on the robot. The robot’s approach motion towards the mouth during the feeding was from the side to avoid co-diner occlusions during the robot movement. This implies the person would have to look away from the co-diners anyway (with or without voice prompting) for the mouth-open condition, which will cause disruption in the social signaling before feeding. During counterbalanced 1-bite per condition study, the user does not know at the beginning of the trial which condition it is going to be, unless they are prompted. Not prompting them would imply the user would have to always keep guessing for which trial to look at the robot, and for which trial not to, and it puts an undue burden on the automated conditions that does not need the user to move their attention away from the conversations before taking the bite. Therefore, prompting at every bite does not put the Mouth-Open baseline at an unfair disadvantage. Instead it is a necessary part of our study design.
> > > > >
> > > > > If the study design was different as the reviewer suggested, then we could do it where the prompting was only at the beginning of the first trial because the user would know that the 5 (or “x”) following trials would be of the same condition. But, we intentionally did not choose that study design because of the reasons mentioned in our earlier posts as well as in the comment below.
> > > > >
> > > > > > "I fully agree that there will be recency biases, and that’s why I suggest randomly shuffling A, B, C (e.g. 5 bites of A, 5 bites of B, 5 bites of C), which should “average out” the recency biases if there is decent population size."
> > > > >
> > > > > The suggested method does not completely remove recency bias/effect, but focuses specifically on removing the ordering effects. Order effects bias the response of the user based on the order of the conditions. Recency bias is a major cognitive bias in human studies, where participants are biased towards the most recently-seen condition because that’s what they remember the most. The longer the time since the last condition, the worse this effect presents itself. With a large number of randomized sessions as the reviewer suggested, only the order effects specifically would be reduced. But, it still does not reduce the recency bias/effect due to the participants struggling to remember what they felt x or more bites ago, especially since each bite occurs between natural and engaging conversations. The larger x is, the more pronounced the effect would be. If the participants cannot clearly remember how the bites ‘x’ or ‘2x’ (x = 5 for example) trials before felt, their preference responses would not be reliable (even if they are counterbalanced to remove ordering effects) as participants won’t remember what to compare the most recent condition against. This is a very widely used rationale and practice in human-robot interaction studies.
> > > > >
> > > > > Additionally, rating the same bite x times in a row would introduce order biases in the per-bite Likert responses. We noticed this empirically during preliminary testing of study designs. Even if we did try to mitigate these issues with counterbalancing (as we discussed in our previous reply), people would still struggle to remember what they felt ‘x’ or ‘2x’ bites ago.

---

> > > > > ### Author Response · Authors · 2022-08-27
> > > > > **Further Discussion with Reviewer iJwW (2/2)**
> > > > >
> > > > > > "I would still like to see some new experimental results while voice prompting is removed."
> > > > >
> > > > > It would be impossible for us to perform this study with a real robot and present these results before the rebuttal period ends. If we ignore the biases a study design with one-time prompting would introduce (as discussed in previous responses) and include this additional condition for evaluation, we would have to recreate the robot study with all the conditions again and ask 30 participants (10 groups) to compare the conditions. Also, as evident from our pilot studies, we are confident that following our study design without prompting will end up creating more confusion among users as to which trials they would have to open their mouth for, and for which they do not. This may end up in more social distraction across all the conditions. If the reviewer thinks that evaluation with a new condition is needed, we can try to achieve this before the final paper is due, but we won't be able to show the results in the remaining rebuttal time.
> > > > >
> > > > > We find this discussion with the reviewer to be interesting and productive, and will be sure to expand our thoughts on experimental design, user-triggered conditions, and disruptions in the appendix.

---

> > > > > > ### Comment · Reviewer_iJwW · 2022-08-28
> > > > > > **Final comment**
> > > > > >
> > > > > > Dear authors,
> > > > > >
> > > > > > Thank you for your detailed response. I'm a lot more convinced about your experimental design now.
> > > > > >
> > > > > > With the authors' comments, I'm happy to change my recommendation to weak accept.

---

> > > ### Author Response · Authors · 2022-08-26
> > > **Discussion with Reviewer iJwW (2/2)**
> > >
> > > > I think getting rid of the voice prompting will make the mouth-open condition closer to real-world applications. Intuitively, voice prompting is the main culprit of creating social distractions, not the "mouth-opening" or "looking-at-the-camera" part.
> > >
> > > Thank you for raising this insightful point. Prompting for every mouth-open condition is necessary for our study design as explained in the comment above. Although we acknowledge that there could be various interfaces that one could use for prompting the user and voice prompting is just one of them, analyzing the effect of interfaces was not the focus of this work. While the choice of the voice prompting interface itself may have created social distractions, using other interfaces such as a silent web-based interface for prompting, or a silent light-based visual interface would also create distractions by focussing the user’s attention on those interfaces and breaking eye contact with other co-diners. There are pros and cons to various interfaces, and while we agree that this would be a very interesting study to conduct and find out how choices of different interfaces affect social distractions, it was not the focus of our work. Some user-triggered interfaces may be perceived as more disruptive than others, but there still needs to be some kind of intervention, which will inevitably cause disruption in natural conversations during social dining, irrespective of the interface used for prompting. We will clarify this in the appendix.

---

> ### Author Response · Authors · 2022-08-25
> **Individual Response to Reviewer iJwW (2/2)**
>
> **R-iJwW-3 >**  _It seems that the model is mostly relying on the bite features from the target user alone to make predictions. It’s a bit unclear whether the gaze, head pose, body, and face features of the diners actually help the model._
>
> The reviewer raises an interesting point. Bite features are indeed the most important as is evident from the ablation studies. Intuitively, it makes sense that bite features are the dominating factor since when someone is eating alone, this is likely the only feature that matters. In a social setting, it follows that it will tend to dominate while other social factors could be important. In these settings, bite features could also encode information from other features, since all features revolve around the act of eating together. We find the trend that all these features do provide a small performance gain in nMCC and accuracy. The small differences in performance do matter, and could possibly change a user’s experience. We have revised our wording to state that we believe body/gaze features are important based on these small differences, though bite features certainly dominate.
>
> **Summary Of Recommendation >** _... it’s still unclear whether predicting bite timing is the key and whether a learned method necessarily is preferred over a much simpler heuristics-based method or a user-triggered mechanism._
>
> Thank you for raising this question. As we have discussed in previous responses, our baselines are real-world user-inspired baselines (which we came up with after discussing with the stakeholders) and hence, are strong and fair baselines. We performed a real-robot user study with these baselines and clearly found that people have a preference towards our learned model using a forced-choice questionnaire. For example, users preferred the learned model 21 times out of 30 times, when compared with the fixed-time interval baseline. For the mouth-open condition baseline, even with full user control on deciding the appropriate bite timing, users felt that it created social distractions during conversations, with statistically significant results against the baseline. We believe that as our robot increases the variety of food items it feeds, the preference for a learned condition would be even more pronounced when compared with the fixed-interval condition, as different food items may need different times for chewing for people with different severity of mobility limitations.
>
> There are other heuristics-based and user-triggered mechanisms that could be considered beyond the ones we chose for our study. However, there are pros and cons to these methods. User-triggered mechanisms could be preferred largely due to the complete control of the system, but it could be inconvenient in social situations or challenging for users with cognitive disabilities. Fixed-interval conditions are fully automated but ignore the dynamic social cues and user preferences that affect bite timing. We believe that our learned condition strikes a balance between these conditions by considering social cues and user preferences while demanding reduced cognitive workload on the user.

---

### Official Review · Reviewer_tBiN · 2022-07-31

**Originality:** Fair
**Technical Quality:** Good
**Clarity Of Presentation:** Good
**Impact:** 3

**Recommendation:**

Weak Accept: I recommend accepting the paper, but will not argue for my recommendation if the majority of other reviewers have a different opinion.

**Summary:**

The paper proposes a method for autonomous assistive feeding, where a robotic arm is able to feed its patient.
In particular, they propose SOcial Nibbling NETwork (SoNNET), a model that predicts when a user has the intention to eat based on various social signals. Experiments include evaluation on their novel dataset with human participants participating in a social meal with 2 other people, in addition to a simplified robot evaluation with the robot feeding fruit to the human participant.
Overall, the main contributions of the paper are the model which captures the subtle interpersonal social dynamics, in human-human and human-robot groups for predicting bite timing, a socially-aware robot-assisted feeding system, and a novel Human-Human Commensality Dataset (HHCD) containing multi-view RGBD video and directional audio recordings capturing 30 groups of three people sharing a meal.


**Issues:**

- The abstract is slightly confusing. I would encourage the authors to start from motivating why their problem is interesting to the robotics community, what they did and finally how, i.e. the opening sentence of the abstract is too abrupt.
- I am not sure it was necessary to say that the people in the collected dataset were fully-vaccinated.
- I understand that for simplicity fruit were chosen, but what was the motivation for this? Also, did you explore other food and in that case what happened? I am also assuming that for the recorded dataset a vast array of food could be used and different cutlery as well so was the robot experiment simplified as a matter of safety? It would be helpful to the research community if you shed some light on your experience with different food/cutlery for this task.
- How did you ensure overall safety with the robot? Could you elaborate further, maybe in the appendix, the control and planning method used on the kinova platform?
- The authors mention the challenge with the dataset in terms of class imbalance. Did you explore strategies to mitigate this for e.g. a weighted loss to take this into account?

I would also encourage the authors to address the weaknesses of the paper to improve it.

**Quality Of The Limitations Section:**

Additional details required

**Reviewer Expertise:**

3: The reviewer is fairly confident that the evaluation is correct

**Robotics Focus:**

Sufficient demonstration on hardware

**Strengths And Weaknesses:**

Strengths
- The authors aim to address an important task, that of assistive feeding, that should it be successful has tremendous societal impact to help people with mobility challenges.
- The ablation study is very interesting. Particularly, they note that "users’ bite features such as the time since last bite and the number of bites are important, as accuracy drops drastically without them. Intuitively, we believe this feature is important because a user’s bite features are a proxy for their level of hunger."
- The authors propose a very interesting solution, where they not only propose a new model but also collect a novel dataset and deploy their proposed methodology on a real robotic platform.
- The paper is well-written and easy to understand and follow.
- Overall, the paper contains a range of practical insights that will be useful for robot learning practitioners.

Weaknesses
- The authors seek to answer the following question: "How should an assistive feeding robot decide the right timing for feeding a user in everchanging and dynamic social dining scenarios?" In my opinion, from their proposed methodology, this hypothesis is not fully met. In this question, they key words are "everchanging and dynamic social dining scenarios". In their model training/evaluation, they carry out their experiments in an isolated room, without environment noise, without moving people in the background as you would expect in social dining scenarios. Hence, their environment is not really representative on what they would like to achieve. I understand that this could be due to facilitation of data collection, and it might also be representative of for example a feeding scenario in a nursing home, but I hope the authors would agree with me that in social scenarios there would be more external disturbances.
- The authors base their model on predicting the timing of a user to take a bite of food based
on the social cues within the interaction. I am wondering if they have any thoughts on whether this would change in different cultures? For e.g. different cultures would have different societal norms (think Mediterranean vs Asian population).
- The authors show that their approach achieves >70% accuracy w.r.t. the recorded dataset. However, they did not present any quantitative results in terms of accuracy w.r.t. the real robot experiment. Would you be able to quantify this for e.g. failure cases = the robot dropped the food, or the robot fed the person but the food fell out. Also, regarding failure cases, could you highlight some examples when it failed and why?
- In the assessment, the authors point out that they understand the limitation of their work is the fact that the used healthy participants. I was wondering how confident they are that their model would work to people with mobility issues? In the video, it is evident that the healthy subject moves closer to the robotic arm for feeding and I am sure there might be cases where this is not possible for humans with severe mobility issues.


**Summary Of Recommendation:**

The paper provides an interesting pipeline for an application with high societal impact, but lacks generalisation across different foods and does not seem to take into account safety, for e.g. if the person moves forward and the robot hits the person.

---

> ### Author Response · Authors · 2022-08-25
> **Individual Response to Reviewer tBiN (1/5)**
>
> We thank the reviewer for a very nice summary of our work and for all the insightful comments. We are glad that the reviewer appreciates the insights within our work to roboticists. Bringing robot-assisted feeding into the social domain comes with its own set of intriguing challenges. We focused on improving bite timing in social settings by constructing a large human-human dataset, learning novel models on the dataset, and then transferring the model onto a real robot. We hope this work motivates considering social group dynamics in other robot learning and assistive tasks.
>
> **R-tBiN-1 >** _In this question, they key words are "everchanging and dynamic social dining scenarios". In their model training/evaluation, they carry out their experiments in an isolated room, without environment noise, without moving people in the background as you would expect in social dining scenarios. I understand that this could be due to facilitation of data collection, and it might also be representative of for example a feeding scenario in a nursing home, but I hope the authors would agree with me that in social scenarios there would be more external disturbances._
>
> Thank you for raising this valid point. We completely agree that in real-life social-dining scenarios, there will be other external disturbances such as people moving in the background. What we meant by “everchanging and dynamic” social dining is that the social dining activity itself is dynamic where people are eating at their own pace, with different gestures and movements, and having natural, unscripted conversations that are not constant over time. During social dining, everything is socially-contextual and the social cues from the participants change and adapt during conversations. In this paper, we do not consider disturbances external to the dining activity. We assume that we have access to a good perception system that can isolate the dining activity from background noise, but this work does not focus on this perception. The creation of these perception algorithms are definitely very interesting, but not within the scope of this work. We will revise the language of the paper to reflect this.
>
> **R-tBiN-2 >** _I am wondering if they have any thoughts on whether this would change in different cultures?_
>
> We believe that culture plays a major role in these scenarios. Social science literature on commensality studies the interplay between social context [1], culture [2, 3], age [4], gender, and various other factors on how long eating takes, what people are eating, etc. Although we have found no studies on bite timing in commensality, it would make sense if these factors also play a role in bite timing. We are excited about the potential of the thorough demographic statistics in our human-human commensality data to investigate the relationship between commensality and culture.
>
> Research in human-robot interaction has generally found that culture plays a role in how humans perceive, trust, and interact with robots [5, 6]. It is likely that the way people react to the presence of a robot can differ across cultures and can change social behaviors. This opens up a new, exciting area of research.
>
> [1] M. Morrison. Sharing food at home and school: perspectives on commensality. The Sociological Review. 1996.
>
> [2] G. Dansei. Commensality in French and German young adults: An ethnographic study. Hospitality & Society. 2012.
>
> [3] C. Fischler. Commensality, society, and culture. Social Science Information. 2011.
>
> [4] S. Biggs and I. Haapala. Intergenerational Commensality: A Critical Discussion on Non-Familial Age Groups Eating Together. Int. J. Environ. Res. Public Health. 2021.
>
> [5] L. Wang, P. -L. P. Rau, V. Evers, B. K. Robinson and P. Hinds, When in Rome: The role of culture & context in adherence to robot recommendations. In 2010 5th ACM/IEEE International Conference on Human-Robot Interaction (HRI), IEEE, 2010.
>
> [6] Lim, V., Rooksby, M. & Cross, E.S. Social Robots on a Global Stage: Establishing a Role for Culture During Human–Robot Interaction. In Int J of Soc Robotics. 2021.

---

> ### Author Response · Authors · 2022-08-25
> **Individual Response to Reviewer tBiN (2/5)**
>
> **R-tBiN-3 >** _The authors show that their approach achieves >70% accuracy w.r.t. the recorded dataset. However, they did not present any quantitative results in terms of accuracy w.r.t. the real robot experiment. Would you be able to quantify this for e.g. failure cases = the robot dropped the food, or the robot fed the person but the food fell out. Also, regarding failure cases, could you highlight some examples when it failed and why?_
>
> We appreciate the reviewer's remarks on the importance of failures in our study. Our study was focused solely on the timing of when the robot should feed a user in a social-dining setting once the robot had picked up the food. Therefore, failures of the robot dropping the food during acquisition are not considered in the timing. Additionally, the robot did not drop the food during the study.
>
> It is difficult to assess “failures” of bite timing in a real robot experiment. Our human-human commensality data has an annotated ground truth value to evaluate against; however, during evaluation time in a real-robot social dining study, these annotations do not exist. Our user study focuses on subjective metrics to determine whether the user is satisfied with their meal experience. After discussing with care recipients, caregivers, and occupational therapists, we realized that instead of relying on bite-timing accuracy, the more relevant metric is whether the users felt rushed or delayed during feeding. None of these can be considered as failures, even though they may not be ideal. It is also not clear what 90% (just an example) accuracy would mean for user satisfaction during feeding, especially when it is unclear as to what a comparable ground-truth would be in human-robot commensality to calculate this accuracy. Because of this, we collected post-trial survey responses to find which methods the user found distracting and which one they preferred. The forced-choice preference question did find that our model was better, but it is difficult to attribute why in terms of user-perceived failures. We, however, agree that failures would play a huge role in determining the overall technology acceptance of a robot-assisted feeding system (as pointed in a related work [1]) but the acceptance questions are not very relevant for this user study that focuses on bite-timing alone and also with users with no mobility limitations.
>
> [1] T. Bhattacharjee, E. K. Gordon, R. Scalise, M. E. Cabrera, A. Caspi, M. Cakmak, and S. S. Srinivasa. Is more autonomy always better? exploring preferences of users with mobility impairments in robot-assisted feeding. In 2020 15th ACM/IEEE International Conference on Human-Robot Interaction (HRI), pages 181–190. IEEE, 2020

---

> ### Author Response · Authors · 2022-08-25
> **Individual Response to Reviewer tBiN (3/5)**
>
> **R-tBiN-4.1 >** _I was wondering how confident they are that their model would work for people with mobility issues?_
>
> We are pasting our response to the meta-reviewer here. We acknowledge that we have not performed our real-robot studies with people with disabilities. This is primarily due to lack of access to stakeholders (care recipients and their caregivers) for this social dining study while following COVID protocols mandated by our university Institutional Review Board (IRB). However, the algorithmic and experimental design decisions were taken by consulting with care-recipients, caregivers, and occupational therapists that our group has established connections with, through years of experience working in this area.
>
> Our target users are people with C3-C5 Spinal Cord Injury in a social dining scenario. Although the target diner is not a person with such an injury, our model is not using their movements to infer when to feed. Instead, our model looks at the motion of the other two diners. We neither observe nor foresee any changes in their behavior when dining with a third person -- with or without this injury. As transferability is a function of their behavior, our experiments demonstrate good transferability across scenarios. We expect it to similarly transfer while dining with a person with the C3-C5 spinal cord injury. People with C3-C5 SCI cannot move their arms to feed themselves due to the constraints in their active range of motion (AROM), but they have some mobility in their neck to lean and take a bite. Neck movement is quite common for feeding such patients and therefore, users in our user study move their neck forward to transfer the food into their mouth. Our Couplet-SoNNET model ignores other features of the user's motion. Our Couplet-SoNNET model produces bite timing based on the other two co-diners, not the user. We have no evidence and/or reason to believe that co-diners’ gestural/behavioral features will significantly change if the user is someone with mobility limitations. This is primarily because in social dining scenarios, the goal is for the care recipient to feel as if they are a part of natural conversations (as pointed out by occupational therapists). Therefore, we expect the model’s inference quality to remain stable. There’s no indication from any of our experiments that domain shift is occurring on the input signal to the model, even across dining instances. People with mobility limitations however may have different preferences, and model personalization could be done on an individual basis.
>
> We would also like to note that in our experiments, the goal of Couplet-SoNNET is to infer when to feed (when to trigger the approach of the robot arm towards the mouth). The actual feeding comes _after_ the decision is made to feed. Therefore, the action of bite transfer into the mouth is a separate component of the experiment. Given these target user-inspired design and algorithmic decisions, we believe our algorithms should be able to generalize to predicting bite-timing for real users. However, we acknowledge that until we perform experiments with real care recipients, we cannot empirically validate that claim. We will add this discussion in the paper to clarify the assumptions and limitations.
>
> **R-tBiN-4.2 >** _In the video, it is evident that the healthy subject moves closer to the robotic arm for feeding and I am sure there might be cases where this is not possible for humans with severe mobility issues._
>
> Our target users are people with C3-C5 Spinal Cord Injury. These people cannot move their arms to feed themselves due to the constraints in their active range of motion (AROM), but they have some mobility in their neck to lean and take a bite. Neck movement is quite common for feeding such patients and therefore, users in our user study move their neck forward to transfer the food into their mouth. However, even for other users with more severe neck mobility limitations, we would like to note that the goal of Couplet-SoNNET is to infer when to feed (when to trigger the approach of the robot arm towards the mouth). The actual feeding comes _after_ the decision is made to feed. Therefore, the action of bite transfer into the mouth is a separate component of the experiment.

---

> ### Author Response · Authors · 2022-08-25
> **Individual Response to Reviewer tBiN (4/5)**
>
> **R-tBiN-5 >** _The abstract is slightly confusing.  I would encourage the authors to start from motivating why their problem is interesting to the robotics community, what they did and finally how, i.e. the opening sentence of the abstract is too abrupt._
>
> We thank the reviewer for their valuable inputs. We were inspired to work on this problem by visiting multiple assistive care facilities and talking to care recipients, caregivers, and occupational therapists. We asked people with mobility limitations what their most memorable meal experiences were, and almost everyone unanimously answered that it was being able to eat with friends and family. We hope this work sets a solid foundation for this line of inquiry and can translate it back into the real world. We will revise the abstract to follow the flow suggested by the reviewer. Thank you again for helping us improve the readability of the manuscript.
>
> **R-tBiN-6 >** _I am not sure it was necessary to say that the people in the collected dataset were fully-vaccinated._
>
> This was a requirement mandated by our university Institutional Review Board (IRB) for this social dining study. This was a social dining study, and eating with others while unmasked posed a danger for spreading the COVID-19 virus. Performing the human-human and human-robot experiments was extremely challenging due to COVID times, especially since these experiments cannot be done under the university IRB while being masked. We had to take special precautions for the IRB approval.
>
> **R-tBiN-7 >** _I understand that for simplicity fruit were chosen, but what was the motivation for this? Also, did you explore other food and in that case what happened? I am also assuming that for the recorded dataset a vast array of food could be used and different cutlery as well so was the robot experiment simplified as a matter of safety?_
>
> Thank you for the insightful comment. Since the focus of our work is on the bite timing problem (not bite acquisition from a plate or a bowl, or bite transfer into the mouth), the acquisition step of the robot during the human-robot commensality experiments was simplified to reduce the potential for failures in robot food acquisition and choosing food items with less chances for food allergies (we sent emails to our participants conforming if they had any allergies associated with the food items). We used easy-to-pick-up fruits (strawberries and cantaloupes). These fruits also were the simplest set of food items that did not have any allergies associated with our participants. We agree that some food items may take significantly longer time to chew which would affect the bite timing, and a model that takes into account the specific food-item may be more robust but that is outside the scope of this paper because of all the infrastructure needed to develop a robot bite acquisition system (which is not the focus of this paper) that can pick up a wide variety of food items with different cutlery such as chopsticks or spoons. We will revise the text of the paper to clarify this.
>
> Note, our human-human commensality dataset (HHCD) was collected with people eating a variety of different meals using various utensils (details in the supplementary material). For the human-human commensality dataset, there were no instructions given to the user as to which food they should eat. They could eat any food they want. So, the resulting HHCD data is very rich with a wide variety of feeding strategies (details in the supplementary material).
>
> **R-tBiN-8.1 >** _How did you ensure overall safety with the robot?_
>
> Thank you for this question. We will expand this section in our appendix. Our robot platform has four levels of safety:
>
> 1. We placed a conservative collision model around the user’s head. The users in our study were familiar with the general workspace of the robot (we moved the robot while they were seated on the chair as a part of the pre-study familiarization procedure).
> 2. The fork has a Force/Torque sensor attached to it, where if a certain threshold of force is reached (beyond acceptable safety / comfortable force thresholds), the arm stops immediately.
> 3. We had an observer watch the experiment while the emergency stop was ready to press in the case of unexpected behaviors. Additionally, an experimenter watched the system and was ready to stop it for safety.
> 4. The compliant robot arm is also set up so that the user can stop it if absolutely necessary. We also designed the speed of the robot to be at comfortable levels.
>
> The participants were made aware of these safety protocols before the experiment, in addition to being familiarized with the workspace of the robot to ensure safety. Of course, any participant could stop the experiment at any point of time if they did not feel it to be safe, but that did not happen for any of our experiments.

---

> ### Author Response · Authors · 2022-08-25
> **Individual Response to Reviewer tBiN (5/5)**
>
> **R-tBiN-8.2 >** _Could you elaborate further, maybe in the appendix, the control and planning method used on the kinova platform?_
>
> For control, we used joint space velocity control. We will add further details in the appendix on the library of planners available to our platform:
> * planToConfiguration(goal_config): plans from current configuration to a joint space goal configuration (6 degrees-of-freedom)
> * planToTaskSpaceRegion(ee_goal_pose, constraints): plans from current configuration to a task space end-effector (EE) goal pose with some given constraints [1]
> * planToEEOffset(ee_offset): plans such that the end-effector moves in the direction of a certain vector.
>
> [1] D. Berenson, S. Srinivasa, J. Kuffner. Task Space Regions: A framework for pose-constrained manipulation planning. The International Journal of Robotics Research. 2011.
>
> **R-tBiN-8.3 >** _The authors mention the challenge with the dataset in terms of class imbalance. Did you explore strategies to mitigate this for e.g. a weighted loss to take this into account?_
>
> During preliminary testing, we did set class weights for positive and negative samples by 1/count, but we did not find any substantial improvements. The imbalance was not extremely large (72% positive - 28% negative) due to evenly distributed negative samples. However, to account for this imbalance during evaluation, we looked at nMCC to provide interpretable metrics.

---

### Official Review · Reviewer_gH2e · 2022-08-05

**Originality:** Good
**Technical Quality:** Good
**Clarity Of Presentation:** Good
**Impact:** 2

**Recommendation:**

Weak Accept: I recommend accepting the paper, but will not argue for my recommendation if the majority of other reviewers have a different opinion.

**Summary:**

In this paper, the authors build a multimodal Human-Human Commensality Dataset and use it to train a SOcial Nibbling NETwork (SoNNET) and use it on a robot-feeding platform for people with mobility limitations. The approach is evaluated both on the dataset and on a real robot with people taking part to the experiment.

**Issues:**

* Related work

**Quality Of The Limitations Section:**

Limitations are addressed clearly

**Reviewer Expertise:**

3: The reviewer is fairly confident that the evaluation is correct

**Robotics Focus:**

Sufficient demonstration on hardware

**Strengths And Weaknesses:**

Strengths:
* The paper is well written and well structured
* The dataset is potentially useful for several applications and is going to be released after acceptance
* The problem is relevant
* The approach is evaluated on the real robot

Weaknesses:
* It would be nice to have a clearer view of what existing approaches there are out there
* While the approach and the problem are relevant, as you state in the limitations, this might not generalize to real people with mobility limitations (and maybe it also depends on the degree of such limitation)

**Summary Of Recommendation:**

The paper is well written and well structured. It deals with a relevant problem that is not considered
sufficiently in robotics, and it does deliver good results, including a publicly released dataset and
several experiments that involve both real people and a real robotic platform. The paper has
obviously some limitations, but I believe it has good quality as well.

It would be great if the authors could contextualize their work a little better within what's the
current status of the research in this field.

---

> ### Author Response · Authors · 2022-08-25
> **Individual Response to Reviewer gH2e (1/2)**
>
> Thank you for your valuable comments. We appreciate that you believe the dataset is going to have several applications. We are particularly excited about the new challenges robot learning practitioners can find and solve using our human-human commensality data. We are also excited about our novel SoNNET models (Triplet-SoNNET for human-human commensality and Couplet-SoNNET for human-robot commensality) that exploit social dynamics to find out appropriate bite timing. We deployed these models on a robot in a real social dining user study, and similarly hope that the community can utilize our algorithms to build new, impactful applications.
>
> **R-gH2e-1** > _It would be nice to have a clearer view of what existing approaches there are out there_
>
> We appreciate the reviewer’s feedback on contextualizing this work to past work. We would like to note that there is no other work that considers the social signals of the other diners to predict bite timing in a social dining study. We were inspired to work on this problem by visiting multiple assistive care facilities and talking to care recipients, caregivers, and occupational therapists. We asked people with mobility limitations what their most memorable meal experiences were, and almost everyone unanimously answered that it was being able to eat with friends and family. We hope this work sets a solid foundation for this line of inquiry and can translate it back into the real world.
>
> This is the first work in assistive feeding that learns from groups to improve feeding in social scenarios. Previous work used the features of only the participant to feed them in a dyadic social scenario using an HMM model [1]. Another work [2] studied interfaces for a dyadic social dining study where the user would either use the interface to indicate they are ready to take a bite, or open their mouth and show their eating intent. This method is exactly one of our baselines used in this paper. Other works in non-social robot-assisted feeding use a mouth-open strategy, a button, or similar manual strategies.
>
> [1] L. Herlant. Algorithms, Implementation, and Studies on Eating with a Shared Control Robot Arm. 2018.
>
> [2] T. Bhattacharjee, E. K. Gordon, R. Scalise, M. E. Cabrera, A. Caspi, M. Cakmak, and S. S. Srinivasa. Is more autonomy always better? exploring preferences of users with mobility impairments in robot-assisted feeding. In 2020 15th ACM/IEEE International Conference on Human-Robot Interaction (HRI), pages 181–190. IEEE, 2020

---

> ### Author Response · Authors · 2022-08-25
> **Individual Response to Reviewer gH2e (2/2)**
>
> **R-gH2e-2 >** _While the approach and the problem are relevant, as you state in the limitations, this might not generalize to real people with mobility limitations_
>
> We are pasting our response to the meta-reviewer here. We acknowledge that we have not performed our real-robot studies with people with disabilities. This is primarily due to lack of access to stakeholders (care recipients and their caregivers) for this social dining study while following COVID protocols mandated by our university Institutional Review Board (IRB). However, the algorithmic and experimental design decisions were taken by consulting with care-recipients, caregivers, and occupational therapists that our group has established connections with, through years of experience working in this area.
>
> Our target users are people with C3-C5 Spinal Cord Injury in a social dining scenario. Although the target diner is not a person with such an injury, our model is not using their movements to infer when to feed. Instead, our model looks at the motion of the other two diners. We neither observe nor foresee any changes in their behavior when dining with a third person -- with or without this injury. As transferability is a function of their behavior, our experiments demonstrate good transferability across scenarios. We expect it to similarly transfer while dining with a person with the C3-C5 spinal cord injury. People with C3-C5 SCI cannot move their arms to feed themselves due to the constraints in their active range of motion (AROM), but they have some mobility in their neck to lean and take a bite. Neck movement is quite common for feeding such patients and therefore, users in our user study move their neck forward to transfer the food into their mouth. Our Couplet-SoNNET model ignores other features of the user's motion. Our Couplet-SoNNET model produces bite timing based on the other two co-diners, not the user. We have no evidence and/or reason to believe that co-diners’ gestural/behavioral features will significantly change if the user is someone with mobility limitations. This is primarily because in social dining scenarios, the goal is for the care recipient to feel as if they are a part of natural conversations (as pointed out by occupational therapists). Therefore, we expect the model’s inference quality to remain stable. There’s no indication from any of our experiments that domain shift is occurring on the input signal to the model, even across dining instances. People with mobility limitations however may have different preferences, and model personalization could be done on an individual basis.
>
> We would also like to note that in our experiments, the goal of Couplet-SoNNET is to infer when to feed (when to trigger the approach of the robot arm towards the mouth). The actual feeding comes _after_ the decision is made to feed. Therefore, the action of bite transfer into the mouth is a separate component of the experiment. Given these target user-inspired design and algorithmic decisions, we believe our algorithms should be able to generalize to predicting bite-timing for real users. However, we acknowledge that until we perform experiments with real care recipients, we cannot empirically validate that claim. We will add this discussion in the paper to clarify the assumptions and limitations.

---

### Meta-Review · Area_Chair_cLBn · 2022-08-14

**Recommendation:** Accept (Poster)
**Confidence:** 4

**Metareview:**

Post discussion update
---------------------------

Thank you to the authors for their thorough answers to my and the reviewers' questions.

The rebuttal raises interesting points about the value of using co-diners social signals only, such as transferability and robustness. I still think a study with the target population is needed to properly gauge satisfaction (the only metric in the real world test, since there is no notion of accuracy, as the rebuttal points out). However, the rebuttal acknowledges this as well, and no paper can be fully complete, so I think in the end it's fine.

Like reviewer iJwW, I'm also not totally satisfied by the explanation for voice prompting in the open-mouth baseline. However, again, I don't think this issue sinks the paper.

Throughout the rebuttal, there is mention of working closely with key stakeholders like people with SCI and clinicians. I would love to see more information about that in the paper, including when those conversations led to certain decisions about the model's implementation or evaluation.

Overall, I believe the main points have been addressed. I'll note that the authors promised to update the following aspects of their paper:
- add more details about the value of using only co-diners' social signals in making the timing decision
- add more details about study design such as the scaling factor in the fixed-time heuristic
- clarify study design decisions for voice prompting in the open-mouth condition
- revise minor language like "hunger"-->"eating rate"


Pre-rebuttal review
----------------

This paper presents an algorithm for robotic eating assistance in social settings. Data about the bite timing from groups of humans was used to train a model that is then applied on a robot. The paper's contributions include a CNN-based architecture, a large hand-annotated dataset, and a user study with able-bodied adults demonstrating the bite timing algorithm.

Strengths
* Reviewers all noted the value of the data set contribution. This work includes an impressively sized data set, with 151 hours annotations from 90 users in groups of 3. Clearly lots of effort was put into this data collection, and it's likely the data will be useable in the future.

* Reviewers agreed that the paper is well written.

* Reviewers agreed that the paper addresses an important research area.

* Reviewers noted that the model worked well compared to strong baselines. The ablation studies revealed interesting information about human-human eating such as the importance of user bite features and the potential social coordination among eaters.


Weaknesses
* Reviewers point out that this approach has not been tested on users with motor impairments, and might not transfer well. Explaining why the results are likely to transfer would be very helpful for justifying the work.

* Reviewers had some critiques about the real-world study. For example, they wanted to know more about results about model accuracy in the real world. They were interested to understand the failure cases of the real world system. They also felt the baseline conditions did not present a fair comparison. Additional information here would strengthen the paper.

* In couplet-SoNNET, social signals from the user are almost entirely ignored. Why? Doesn't it matter, for example, that the user is talking, or looking away, or providing some other social signal that they're not ready for a bite?

* Also, please don't use "healthy" to refer to unimpaired people, as it sets up the notion that people with disabilities are "unhealthy."

* The paper says that bite features are a proxy for hunger. Is this really the case, or are they just a proxy for the rate at which people eat? Is there any evidence that hunger is what drives the bite features?

---

> ### Author Response · Authors · 2022-08-25
> **Response to Meta-Reviewer (1/2)**
>
> We thank the meta-reviewer and the reviewers for their valuable feedback. We are glad that all the reviewers see the impact of studying robot-assisted feeding in social settings and the importance of our large dataset of human-human commensality. In this work, we focus on improving bite timing in social scenarios. We built our novel SoNNET models (Triplet-SoNNET for human-human commensality and Couplet-SoNNET for human-robot commensality) that exploit social dynamics to find out appropriate bite timing. We also evaluated our model in a novel real-robot social dining study. Using our comprehensive dataset and novel learning algorithms that model social dynamics, we are excited for the robot learning community to find other interesting challenges.
>
> We appreciate your constructive feedback on our paper. We address each reviewer comment individually in detail below as separate comments. We also address the meta-reviewer’s feedback, some of which are common across reviewers here:
>
> **MR-1 >** _Reviewers point out that this approach has not been tested on users with motor impairments, and might not transfer well. Explaining why the results are likely to transfer would be very helpful for justifying the work._
>
> We acknowledge that we have not performed our real-robot studies with people with disabilities. This is primarily due to lack of access to stakeholders (care recipients and their caregivers) for this social dining study while following COVID protocols mandated by our university Institutional Review Board (IRB). However, the algorithmic and experimental design decisions were taken by consulting with care-recipients, caregivers, and occupational therapists that our group has established connections with, through years of experience working in this area.
>
> Our target users are people with C3-C5 Spinal Cord Injury in a social dining scenario. Although the target diner is not a person with such an injury, our model is not using their movements to infer when to feed. Instead, our model looks at the motion of the other two diners. We neither observe nor foresee any changes in their behavior when dining with a third person -- with or without this injury. As transferability is a function of their behavior, our experiments demonstrate good transferability across scenarios. We expect it to similarly transfer while dining with a person with the C3-C5 spinal cord injury. People with C3-C5 SCI cannot move their arms to feed themselves due to the constraints in their active range of motion (AROM), but they have some mobility in their neck to lean and take a bite. Neck movement is quite common for feeding such patients and therefore, users in our user study move their neck forward to transfer the food into their mouth. Our Couplet-SoNNET model ignores other features of the user's motion. Our Couplet-SoNNET model produces bite timing based on the other two co-diners, not the user. We have no evidence and/or reason to believe that co-diners’ gestural/behavioral features will significantly change if the user is someone with mobility limitations. This is primarily because in social dining scenarios, the goal is for the care recipient to feel as if they are a part of natural conversations (as pointed out by occupational therapists). Therefore, we expect the model’s inference quality to remain stable. There’s no indication from any of our experiments that domain shift is occurring on the input signal to the model, even across dining instances. People with mobility limitations however may have different preferences, and model personalization could be done on an individual basis.
>
> We would also like to note that in our experiments, the goal of Couplet-SoNNET is to infer when to feed (when to trigger the approach of the robot arm towards the mouth). The actual feeding comes _after_ the decision is made to feed. Therefore, the action of bite transfer into the mouth is a separate component of the experiment. Given these target user-inspired design and algorithmic decisions, we believe our algorithms should be able to generalize to predicting bite-timing for real users. However, we acknowledge that until we perform experiments with real care recipients, we cannot empirically validate that claim. We will add this discussion in the paper to clarify the assumptions and limitations.

---

> ### Author Response · Authors · 2022-08-25
> **Response to Meta-Reviewer (2/2)**
>
> **MR-2 >** _Reviewers had some critiques about the real-world study. For example, they wanted to know more about results about model accuracy in the real world. They were interested in understanding the failure cases of the real world system. They also felt the baseline conditions did not present a fair comparison._
>
> **Reviewer tBiN** wondered about the accuracy of Couplet-SoNNET in the real world. Our HHCD data has an annotated ground truth value to evaluate against; however, during evaluation in a real-robot social dining study, these annotations do not exist. Our user study focuses on subjective metrics to determine whether the user is satisfied with their meal experience. After discussing with care recipients, caregivers, and occupational therapists, we realized that instead of relying on bite-timing accuracy, the more relevant metric is whether the users felt rushed or delayed during feeding. Because, it is not clear what 90% (just an example) accuracy would mean for user satisfaction during feeding, especially when it is unclear as to what a comparable ground-truth would be in human-robot commensality to calculate this accuracy. We discuss this in detail in response to **Reviewer tBiN** in **R-tBiN-3**.
>
> **Reviewer iJwW** had concerns about the baseline conditions, particularly with the Mouth-Open Timing. We designed these baseline conditions by going to assisted care living facilities and discussing with care recipients, occupational therapists, and caregivers how they know when to feed. Caregivers said that they estimate bite timing when care recipients open their mouth and look at them. We believe our baselines are fair and inspired by real-world usage. We provide a lengthy discussion in a response post to **Reviewer iJwW**.
>
> **MR-3 >** _In couplet-SoNNET, social signals from the user are almost entirely ignored. Why? Doesn't it matter, for example, that the user is talking, or looking away, or providing some other social signal that they're not ready for a bite?_
>
> Thank you for raising this point. As discussed in the Couplet-SoNNET subsection in Section 4 of our paper, our HHCD data consists of diners who are eating with their own cutlery. Most of the signals that are indicators of eating or not-eating (such as head pose) during self-feeding are influenced by the fact they are holding a utensil. For a user who is being fed by a robot, we cannot use these features. We gave a lot of thought about our target population. We consulted people with disabilities, caregivers, and occupational therapists on what features we should look at based on what movements are consistent across people with disabilities. Our target users (with C3-C5 SCI) cannot move their arms to feed themselves. There is a huge spectrum of severity of mobility limitations depending on the users’ conditions and their movements are not consistent across these users. Therefore, Couplet-SoNNET uses only the features of the co-diners to make it more generalizable across this target population. We include an image of this model in the attached files, as we only depict Triplet-SoNNET in the paper. We will also revise the text to make this point clearer in the paper.
>
> Whether a user is talking could be relevant to predicting bite timing. During preliminary testing, we found that modeling the user’s speaking status led to the model never feeding at all if they kept talking (since talking is highly correlated with not-feeding in HHCD). Since the user is not self-feeding, they are not incentivized to stop talking. Therefore, we believe some level of coercion is required to ensure the user is fed (which we realize is a common subtle practice, when we spoke with the caregivers who feed care-recipients). By removing the user’s speaking status features, we can ensure that feeding does occur. We will revise the paper’s text to make this point clear.
>
> **MR-4** > _Also, please don't use "healthy" to refer to unimpaired people, as it sets up the notion that people with disabilities are "unhealthy."_
>
> Thank you for pointing out this extremely important error in terminology. We apologize for this oversight. We choose to use people-first language (people with no mobility limitations), rather than identity-first language (able-bodied participants, unimpaired people). People-first language is generally the standard in the United States. We will fix this in the final draft of our paper and will use the term “people with no mobility limitations”.
>
> **MR-5** >_The paper says that bite features are a proxy for hunger. Is this really the case, or are they just a proxy for the rate at which people eat? Is there any evidence that hunger is what drives the bite features?_
>
> We agree with the meta-reviewer that the statement that bite features are a proxy for hunger is too strong. It is likely a proxy for the rate at which people eat. Hunger plays an important role in this, but there could be other factors. We recognize this and will update the language in the paper.